

# Interprofessional collaboration and associated factors among wound, ostomy, and continence nurses: a cross-sectional study in China

Lin Shi, Liqing Yue, Xiaowan Liu, Xueqin Gong and Bingfa Li

Teaching and Research Section of Clinical Nursing, Xiangya Hospital, Central South University, Changsha, China

## ABSTRACT

**Background**. Wound, ostomy, and continence (WOC) nurses play a crucial role in providing cost-effective and high-quality care for an increasing number of people with acute and chronic wounds. WOC nursing is a multifaceted, evidence-based specialty practice requiring inter-professional collaboration (IPC). IPC is a core competency that ensures comprehensive and holistic wound management, yet few studies have examined IPC among WOC nurses in China. This study aimed to investigate the level and influencing factors of IPC among WOC nurses in China, providing insights into IPC education and intervention.

**Methods**. A cross-sectional study was conducted among 247 WOC nurses recruited from various hospitals from January to February 2025. Participants completed an online questionnaire to assess their IPC competency, readiness for interprofessional learning, and work environment. Multiple linear regression was used to determine influencing factors for IPC competency.

**Results**. The mean scores for IPC, readiness for inter-professional learning, and work environment were $134.49 \pm 10.99$, $81.15 \pm 6.00$, and $127.1 \pm 18.78$, respectively. A higher level of inter-professional learning readiness ($\beta = 0.46, P < 0.001$), a better work environment ($\beta = 0.39, P < 0.001$), and being a WOC team leader ($\beta = 0.12, P = 0.008$) were associated with a higher level of IPC competency, collectively explaining 55.1% of the total variance.

**Conclusion**. WOC nurses showed a generally good level of IPC, but there is still room for improvement. IPC may be effectively improved through optimizing the work environment, strengthening IPC learning, and empowering nurses with leadership responsibilities.

Corresponding author
Bingfa Li, lb751109880@163.com

## INTRODUCTION

Chronic wounds, such as pressure injuries, venous ulcers, and diabetic foot ulcers, are a silent epidemic that pose a tremendous and rising clinical, social, and economic burden worldwide (*Graves, Phillips & Harding, 2022*). The global aging populations, increased diabetes prevalence, and ongoing infection challenges have all contributed to the growing

number of patients with chronic wounds (*Sen, 2023*). In China, about 100 million people need wound care every year, and about 30 million face chronic wounds that are difficult to heal (*Wang, 2016*). The increasing number of patients with chronic wounds and the associated high disease burden indicate a rapidly increasing need for professional nursing and care.

As tri-specialty nurses, wound, ostomy, and continence (WOC) nurses play a crucial role in the care management of chronic wounds (*Zhang et al., 2022*). WOC nurses are a group of multidisciplinary specialty nurses with advanced knowledge and skills in the prevention, care, intervention, and rehabilitation of a wide range of chronic wound problems (*Carmel, Colwell & Goldberg, 2021*). In wound care, chronic wound patients often have complex conditions and comorbidities that involve a collaborative team, requiring ongoing development of knowledge, skills, attitudes, values, and judgment for effective team integration (*Canadian Interprofessional Health Collaborative, 2010*). Therefore, inter-professional collaboration (IPC) has become a core competency for WOC nurses to provide comprehensive wound care and management (*Joret et al., 2019*; *Moore et al., 2019*).

IPC is defined as a "partnership between people from diverse backgrounds with distinctive professional cultures and possibly representing different organizations or sectors, who work together to solve problems or provide services" (*Morgan, Pullon & McKinlay, 2015*). IPC has been well recognized as a primary strategy for achieving high-quality holistic care, as it enhances team efficiency, reduces healthcare costs, improves patient outcomes, reduces complications, and lowers mortality while increasing both patient and team satisfaction (*Goldsberry, 2018*; *De Sutter et al., 2019*; *Reeves et al., 2017*). Notably, IPC has been shown to reduce hospital admissions, major amputations, mortality, and treatment costs in managing diabetic foot ulcers (*Joret et al., 2019*).

Empirical evidence of IPC is predominantly derived from Western countries such as North America (*Brewer et al., 2016*), which may have different cultures and health systems from developing countries such as China, indicating a need for a culture-sensitive understanding of IPC across countries (*Hu & Broome, 2020*). China has a unique healthcare system that is totally dependent on the government, and the recent health reforms have attached great importance to integrative and cooperative healthcare, emphasizing IPC (*Li et al., 2017*). In addition, Chinese culture is deeply rooted in collectivism and influenced by Confucian values, emphasizing harmony, conformity, and hierarchy, with group needs prioritized over individual needs (*Li et al., 2019*). These values are also embedded in the Chinese healthcare system and profoundly influence the implementation of IPC (*Hu & Broome, 2020*). Therefore, it is necessary to explore the level of IPC among WOC nurses in a Chinese context.

In China, nurses are confronted with multiple barriers to IPC, which involve the organizational level, inter-individual level, and individual level (*Rawlinson et al., 2021*). For instance, China's healthcare system has a strong hierarchical structure that emphasizes physician authority, which may discourage nurses from active collaboration and shared decision-making (*Shi et al., 2023*). Additionally, China's traditional education system often separates disciplines, with most healthcare programs focusing on individual professional training rather than IPC training, which may lead to a lack of role clarity and poor

communication (*Jiang et al., 2024*). However, the impacts of the work environment and learning ability on WOC nurses' IPC are rarely explored among WOC nurses in China.

In sum, WOC nurses play a crucial role in improving patient outcomes and reducing healthcare costs. IPC is essential in facilitating multidisciplinary team cooperation to provide high-quality WOC care and services. While there are clear post responsibilities and role functions of WOC nurses in developed Western countries, the development and cultivation of IPC in WOC nurses in developing countries, such as China, is relatively new and less studied (*Liu & Wang, 2018*). Therefore, we conducted the current study to assess the level of IPC competency and its influencing factors among Chinese WOC nurses. Specifically, we hypothesized that the work environment and WOC nurses' learning readiness were positively associated with their IPC competency. Our findings will inform educational programs and guide the design of targeted interventions that enhance IPC skills among WOC nurses.

## METHODS

### Study design and participants

From January to February 2025, a cross-sectional study was conducted online through WeChat, China's biggest social media app with a wide range of functions, including text messaging, video conferencing, mobile payment, and sharing of photographs and videos (*Montag, Becker & Gan, 2018*). Due to its popularity and multipurpose applications, WeChat has been widely utilized in the workplace for communication and information sharing. This study used a convenience sampling method to recruit WOC nurses from the National Wound Care Nurse WeChat Group, which includes 500 members from various hospitals at different levels all over the country. Inclusion criteria were as follows: (1) age $\geq$ 18; (2) certified WOC nurses who had completed systematic training for WOC care with certification (such as foreign wound treatment certification, domestic wound-therapist certification, and wound, stoma, and incontinence nursing certification); (3) registered nurses with a valid nursing practice license; (4) with over one year of working experience in WOC; (5) with informed consent to participate in the study. Exclusion criteria included: (1) nurses who were off duty for more than three months due to illness, personal matters, maternity leave, or study; (2) nurses who obtained WOC certification but had not engaged in any clinical work in WOC care. Sample size was calculated according to the form for a cross-sectional study: $n = Z_{1-\alpha/2}^2 SD^2/d^2$. Z was set as 1.96 at a significance level ($\alpha$) of 0.05, SD (the standard deviation of the IPC score) was set at 10 based on our pilot study, and d (absolute error) was set at 1.4. Considering a rejection or loss-to-follow-up rate of 20%, we expanded our final sample size to 235. The final sample included 247 participants, which satisfied the sample size requirement.

### Data collection

The study was approved by the Ethics Committee of Xiangya Hospital of Central South University (No. 2024121785). Participants were recruited online through the National Wound Care Nurse WeChat Group. Prior to the study, an electronic advertising leaflet about this study was distributed to the WeChat group to introduce the research and

encourage participation. Those who were interested in participating in the survey could scan a QR code that was attached to the leaflet, which would bring them to an online questionnaire through SoJump. SoJump is China's largest online platform for research design, survey distribution, data collection, and data analysis (*Ponte et al., 2024*). On the first page of the questionnaire, there was a letter explaining in detail the study's purpose, significance, procedure, benefits, and risks, followed by an electronic informed consent form. After providing informed consent, the participants were asked to complete a battery of questionnaires to collect information on their demographics as well as assess their IPC, work environment, and learning readiness. Each questionnaire has clear guidance and explanation to facilitate participants' answers. The survey was conducted anonymously, and all the responses were kept confidential. To ensure quality, each IP address could submit the questionnaire once, with all questions thoroughly answered. After data collection, two researchers cleaned and analyzed the data, removing invalid questionnaires with repeating answer patterns or submitted within 3 min.

## Measurements

A researcher-designed questionnaire was developed based on a literature review, expert opinions, and clinical experience. The questionnaire first collected basic information about the participants, including gender, age, specialty working years, education, professional title, administrative position, only child, marital status, hospital level, off-site learning experience in the past three years, employment mode, work pattern, and whether serving as a specialty team leader. In China, higher-level degrees are classified into four levels: associate's degree (obtained through a vocational program at a junior college), bachelor's degrees, master's degrees, and doctoral degrees (*Gu, Li & Wang, 2018*). For this study, we combined master's and doctoral degrees into one category due to the small number in each category. Chinese hospitals are categorized into three levels: primary (Level I), secondary (Level II), and tertiary (Level III), with primary hospitals mainly providing community-based general care, secondary hospitals providing comprehensive health services for a region, and tertiary hospitals providing specialist care for multiple urban areas (*Zhou et al., 2021*). In addition, we assessed IPC, readiness for inter-professional learning, and nursing work environment using standard scales.

## Chiba inter-professional competency scale

The Chiba inter-professional competency scale (CICS29) is a multi-dimensional self-report instrument assessing the degree of collaborative competencies among various healthcare providers (*Sakai et al., 2017*). It was initially developed by *Sakai et al. (2017)* and translated into Chinese by *Huang (2020)*. It consists of 29 items under six domains: attitudes and beliefs as a professional (six items), team management skills (five items), actions for accomplishing team goals (five items), providing care that respects patients (five items), attitudes and behaviors that improve team cohesion (four items), and fulfilling one's role as a professional (four items). Each item is rated on a 5-point Likert scale ranging from 1 (strongly disagree) to 5 (strongly agree). The total score ranges from 29 to 145, with a higher score indicating a higher level of IPC competency. The Chinese version of CICS29 showed good internal consistency with a Cronbach's α coefficient of 0.996.

## Nursing work environment scale

The Nursing work environment scale (NWES) is a multi-dimensional self-report instrument developed by *Shao, Ye & Tang (2016)* to assess nurses' perceptions of their work environment. It includes 26 items under seven domains: career development (five items), leadership and management (four items), doctor-nurse relationship (four items), recognition (three items), professional autonomy (four items), basic security (three items), and adequate staffing (three items). Each item is rated on a 6-point Likert scale ranging from 1 (strongly disagree) to 6 (strongly agree). The total score ranges from 26 to 156, with a higher score indicating a better perceived nursing work environment. The NWES showed good internal consistency with a Cronbach's α coefficient of 0.962.

## Readiness for Inter-professional Learning Scale

The Readiness for Inter-professional Learning Scale (RIPLS) is a multi-dimensional self-report instrument evaluating learners' readiness in teamwork, professional identity, and role responsibilities. It was initially developed by *Mahler, Berger & Reeves (2015)* and translated into Chinese by *Wang & Hu (2017)*. It contains 19 items under four domains: teamwork and collaboration (items one to nine), negative professional identity (items 10 to 12), positive professional identity (items 13 to 16), and role and responsibilities (items 17 to 19). Each item is rated on a 5-point Likert scale ranging from 1 (strongly disagree) to 5 (strongly agree) except for the items of negative professional identity, which were phrased negatively and reverse-scored. The total score ranges from 19 to 95, with a higher score indicating better preparedness for IPC learning. The Chinese version of RIPLS showed good internal consistency with a Cronbach's α coefficient of 0.887.

## Statistical methods

Statistical analysis was performed using SPSS 23.0 (IBM Corp., Armonk, NY, USA). For descriptive analysis, categorical variables were expressed as frequencies and proportions, while continuous variables were expressed as means and standard deviations. The scores of CICS29, NWES, and RIPLS were further transformed into scoring rates by dividing the actual score of each scale by the total possible score for that scale and then multiplying by 100 to express as a percentage (*Soemantri et al., 2022*). For quantitative variables, this study categorized age and years of work experience into five-year intervals to investigate whether there are differences in the IPC competency of WOC nurses across different groups. Pearson correlation was used to analyze relationships between IPC, work environment, and readiness for inter-professional learning. One-way ANOVA and independent sample t-tests were used to assess differences in IPC scores across various characteristics, and multiple linear regression was used to determine influencing factors for IPC. The significance level was set at $\alpha = 0.05$.

## RESULTS

### Descriptive analysis

A total of 270 questionnaires were collected, with 247 valid responses after excluding those completed in less than 3 min or containing duplicate answers, yielding a 91.48% valid

**Table 1   Scores of CICS29, NWES, and RIPLS ($n = 247$).**

| Variable | Dimension score $\bar{x} \pm s$ | Entry score $\bar{x} \pm s$ |
|---|---|---|
| **Total CICS29** | 134.49 ± 10.99 | 4.64 ± 0.38 |
| Attitudes and beliefs as a professional | 28.13 ± 2.22 | 24.69 ± 0.37 |
| Team-management skills | 22.74 ± 2.24 | 64.55 ± 0.45 |
| Actions for accomplishing team goals | 22.83 ± 2.67 | 54.57 ± 0.53 |
| Providing care that respects patients | 23.84 ± 1.87 | 14.77 ± 0.37 |
| Attitudes and behaviors that improve team cohesion | 18.38 ± 2.01 | 44.60 ± 0.50 |
| Fulfilling one's role as a professional | 18.57 ± 1.86 | 34.64 ± 0.46 |
| **Total NWES** | 127.1 ± 18.78 | 4.88 ± 0.72 |
| Career development | 23.87 ± 4.98 | 4.77 ± 1.00 |
| Leadership and management | 18.85 ± 3.78 | 4.71 ± 0.94 |
| Doctor-nurse relationship | 20.46 ± 4.07 | 5.12 ± 0.79 |
| Recognition | 16.15 ± 2.03 | 5.38 ± 0.68 |
| Professional autonomy | 20.82 ± 2.92 | 5.20 ± 0.73 |
| Basic security | 13.46 ± 3.40 | 4.49 ± 1.13 |
| Adequate staffing | 13.46 ± 3.08 | 4.49 ± 1.03 |
| **Total RIPLS** | 81.15 ± 6.00 | 4.01 ± 0.98 |
| Team and collaboration | 42.68 ± 3.49 | 4.74 ± 0.39 |
| Negative professional identity | 12.98 ± 1.76 | 4.33 ± 0.59 |
| Positive professional identity | 18.48 ± 2.00 | 4.62 ± 0.50 |
| Role and responsibility | 7.01 ± 2.28 | 2.34 ± 0.76 |

Notes.
CICS29, Chiba Inter-professional competency scale; NWES, Nursing work environment scale; RIPLS, Readiness for Inter-professional Learning Scale.

response rate. The mean scores for the CICS29, NWES, and RIPLS were 134.49 ± 10.99 (out of 145, scoring rate: 92.75%), 127.10 ± 18.78 (out of 156, scoring rate: 81.47%), and 81.15 ± 6.00 (out of 95, scoring rate: 85.42%), respectively. Detailed scores for each dimension are presented in Table 1.

## Comparison of IPC scores by sample characteristics

Table 2 shows the comparison of IPC competency by various sample characteristics. The results showed significantly higher scores of IPC competency in those who had off-site learning experiences in the past three years (137.46 ± 8.79 *vs.* 132.69 ± 11.76, $P < 0.001$), served as a nursing manager (135.91 ± 10.14 *vs.* 132.09 ± 11.90, $P = 0.011$), and served as a specialty team leader (136.36 ± 9.65 *vs.*.133.26 ± 11.62, $P = 0.017$).

## Correlation analysis between key variables

Table 3 shows the Pearson correlation analysis among CICS29, NWES, and RIPLS scores. The CICS29 score was significantly and positively correlated with NWES ($r = 0.57$, $P < 0.001$) and RIPLS scores ($r = 0.54$, $P < 0.001$). The NWES score was positively correlated with the RIPLS ($r = 0.38$, $P < 0.001$).

**Table 2 Comparison of CICS29 by different sample characteristics (*n* = 247).**

| Variable | Frequency (%) | Total scores Mean ± SD | t/F | *p*-value |
|---|---|---|---|---|
| **Gender** | | | $t = -1.187$ | 0.236 |
| Male | 11 (4.5) | 130.64 ± 11.66 | | |
| Female | 236 (95.5) | 134.67 ± 10.92 | | |
| **Age (years)** | | | $F = 2.011$ | 0.570 |
| ≤35 | 79 (32%) | 133.63 ± 11.39 | | |
| 36 to 40 | 83 (33.6%) | 134.16 ± 10.83 | | |
| 41 to 45 | 40 (16.2%) | 135.55 ± 11.40 | | |
| ≥45 | 45 (18.2%) | 135.64 ± 9.95 | | |
| **Specialty working years** | | | $F = 0.340$ | 2.157 |
| ≤5 | 123 (49.8%) | 133.26 ± 11.79 | | |
| 6 to 10 | 82 (33.2%) | 135.11 ± 10.70 | | |
| ≥11 | 42 (17%) | 136.86 ± 8.28 | | |
| **Education** | | | $F = 0.900$ | 0.637 |
| Associate's degree | 10 (4%) | 133.60 ± 10.83 | | |
| Bachelor's degree | 222 (89.9%) | 134.71 ± 10.94 | | |
| Master's degree | 15 (6.1%) | 131.8 ± 11.44 | | |
| **Professional title** | | | $F = 0.095$ | 0.136 |
| Primary | 17 (6.9%) | 133.18 ± 10.93 | | |
| Intermediate | 168 (68%) | 133.99 ± 11.06 | | |
| Senior | 62 (25.1%) | 136.19 ± 10.61 | | |
| **Administrative position** | | | $t = -3.153$ | <0.001 |
| No | 154 (62.3%) | 132.69 ± 11.76 | | |
| Nursing manager | 93 (37.7%) | 137.46 ± 8.79 | | |
| **Only child** | | | $t = 2.262$ | 0.145 |
| Yes | 48 (19.4%) | 132.75 ± 10.61 | | |
| No | 199 (80.6%) | 134.90 ± 10.51 | | |
| **Marital status** | | | $t = 1.491$ | 0.137 |
| Married | 229 (92.7%) | 134.90 ± 10.71 | | |
| Unmarried | 18 (7.3%) | 129.22 ± 12.95 | | |
| **Hospital level** | | | $F = 0.031$ | 0.628 |
| Tertiary hospital | 173 (70.1%) | 134.09 ± 11.28 | | |
| Secondary hospital | 29 (11.7%) | 135.34 ± 9.83 | | |
| Primary hospital | 45 (18.2%) | 135.44 ± 10.43 | | |
| **Off-site learning experience in the past three years** | | | $t = -2.946$ | 0.011 |
| No | 92 (37.2%) | 132.09 ± 11.90 | | |
| Yes | 155 (62.8%) | 135.91 ± 10.14 | | |
| **Employment mode** | | | $t = 0.311$ | 0.756 |
| Authorized | 148 (59.9%) | 135.12 ± 10.67 | | |
| Contract | 99 (40.1%) | 133.54 ± 11.37 | | |
**Table 2** (*continued*)

| Variable | Frequency (%) | Total scores Mean ± SD | t/F | p-value |
|---|---|---|---|---|
| **Work pattern** | | | $F = -0.047$ | 0.460 |
| Full-time | 75 (30.4%) | 135.45 ± 10.41 | | |
| Part-time | 172 (69.6%) | 134.06 ± 11.20 | | |
| **Serving as a specialty team leader** | | | $t = 2.197$ | 0.017 |
| Yes | 98 (39.7%) | 136.36 ± 9.65 | | |
| No | 149 (60.3%) | 133.26 ± 11.62 | | |

Notes.
    CICS29, Chiba Inter-professional Competency scale.

**Table 3 Correlation analysis among CICS29, NWES, and RIPLS.**

| | M ± SD | CICS29 | NWES | RIPLS |
|---|---|---|---|---|
| **CICS29** | 134.49 ± 10.99 | 1 | | |
| **NWES** | 127.1 ± 18.78 | 0.574*** | 1 | |
| **RIPLS** | 81.15 ± 6.00 | 0.544*** | 0.378*** | 1 |

Notes.
*** $P < 0.001$
    CICS29, Chiba Inter-professional competency scale; NWES, Nursing work environment scale; RIPLS, Readiness for Inter-professional Learning Scale.

**Table 4 Multiple linear regression analysis of influencing factors of the CICS29 score ($n = 247$).**

| Variable | B | Std. Error | β | T | P |
|---|---|---|---|---|---|
| Constant term | 38.21 | 6.914 | - | 5.527 | <0.001 |
| NWES | 0.23 | 0.028 | 0.39 | 11.172 | <0.001 |
| RIPLS | 0.84 | 0.087 | 0.46 | 9.659 | <0.001 |
| Serving as a specialty team leader | 2.76 | 1.035 | 0.12 | 2.669 | 0.008 |

Notes.
    NWES, Nursing work environment scale; RIPLS, Readiness for Inter-professional Learning Scale.

## Multiple linear regression analysis of influencing factors of IPC competency

A multiple linear regression analysis was conducted to explore the influencing factors of IPC competency with the total CICS29 as the dependent variable. Independent variables included five statistically significant variables from univariate and correlation analyses: nursing management role, specialty team leadership, off-site learning experiences in the past three years, nursing work environment, and readiness for inter-professional learning. Collinearity diagnostics showed tolerance values below 1 and VIF values between 1.056 and 1.258, indicating no multi-collinearity among the independent variables. The results showed that a higher level of inter-professional learning readiness ($\beta = 0.46$, $P < 0.001$), a better work environment ($\beta = 0.39$, $P < 0.001$), and being a WOC team leader ($\beta = 0.12$, $P = 0.008$) were associated with a higher level of IPC competency, collectively explaining 55.1% of the total. Details are shown in Table 4.

## DISCUSSION

### General IPC level

Our study showed that the average IPC competency score among WOC nurses in China was 134.49 ± 10.99 (scoring rate: 92.75%), which was higher than the reported 128.53 (scoring rate: 88.6%) among healthcare professionals in Indonesia (*Soemantri et al., 2022*). The level of IPC competency in our study was higher than in most previous studies conducted in other countries using various assessment tools, such as Ethiopia (*Degu et al., 2023*), Italy (*Rapetti et al., 2014*), and Egypt (*Mohamed & Hasanin, 2021*). It was also higher than those reported in East Asian countries with similar cultural and organizational systems, such as Korea (*Kim et al., 2022*) and Japan (*Haruta, Ozone & Goto, 2019*). The relatively high level of IPC competency in our study may be related to the high importance attached to the development and advancement of WOC nurses in China. WOC nurses regularly collaborate with professionals from multiple disciplines, such as orthopedics, dermatology, and vascular surgery. This interaction is part of the multidisciplinary approach to enhance WOC nurses' IPC skills in wound management across hospitals and community care centers (*Hu et al., 2023*).

Among the six dimensions of the IPC competency scale, the dimension of providing care that respects patients scored the highest, which was consistent with *Li et al. (2023)* study in China using the same scale. These findings reflect nurses' strong emphasis and commitment to providing patient-centered care services in cross-professional collaboration. However, it should be noted that WOC nurses in our study scored lowest in the dimension of team management skills. The Chinese legal framework focuses on a hierarchical nursing management system, where nurses are in a subordinate position (*Wang et al., 2024*). In addition, the education programs on nursing also attach more importance to practical nursing skill training than leadership and teamwork development, leading to a lack of knowledge and low empowerment for nurses to engage in team management (*Jiang et al., 2024*). Furthermore, high job demands and burnout can significantly impact a nurse's ability to effectively collaborate and manage within a team, leading to low scores in team management skills (*Niinihuhta & Haggman-Laitila, 2022*). Our findings stress the importance of strengthening IPC education and training among WOC nurses, with a special focus on improving their team management skills.

### Influencing factors of IPC competency
#### Nursing work environment

Our study showed that the average total score of the nursing work environment was 127.1 ± 18.78 (scoring rate: 81.47%), indicating that the nursing work environment was at a medium level. Among the seven domains, WOC nurses scored lowest in the domains of adequate staffing and basic security. Inadequate staffing leads to increased workload for nurses, which will increase the risk of medical errors and nurse injuries, as well as increased nurse burnout and job dissatisfaction (*Nantsupawat et al., 2022*). Nurses face a higher risk of workplace violence compared to other industries, which not only impacts the nurses' safety and mental health but also compromises the quality of patient care (*Mohammadi Gorji et al., 2021*).

In addition, a better-perceived nursing work environment by Wound, Ostomy, and Continence Nurse Society (WOCN) nurses was associated with a higher level of IPC competency, with a medium effect size ($\beta = 0.39$, $P < 0.001$). Such an effect size has significant practical significance, as this finding implies that interventions aimed at improving the WOCN work environment could lead to better IPC levels among these nurses. These findings were consistent with the international literature showing a positive association between nursing work environment and IPC abilities across various populations (*e.g.*, nurses and students) and in multiple countries (*e.g.*, Oman, Peru) (*Labrague et al., 2022*; *Berduzco-Torres et al., 2020*). Therefore, nursing managers should improve organizational policies and form a supportive teamwork atmosphere to ensure the scientific and accurate allocation of human and material resources, thus promoting high-quality IPC among WOC nurses.

### Readiness for inter-professional learning

Our study found that the average total score of inter-professional learning readiness was $81.15 \pm 6.00$ (scoring rate: 85.42%), suggesting that WOC nurses in China generally have a positive attitude towards inter-professional cooperative learning. Among the four domains, WOC nurses scored lowest in the domain of role responsibility, indicating a lack of self-awareness of their professional roles and functions. Therefore, nurse educators and mentors should help to instill a greater sense of professionalism and activism in nurses, emphasizing their vital role in shaping the future of healthcare.

In addition, WOC nurses with a higher level of inter-professional learning readiness had higher IPC competency, with a medium effect size ($\beta = 0.46$, $P < 0.001$). The medium effect size observed in this study suggests that the improvement in inter-professional learning readiness can have a tangible and meaningful impact on IPC competency, highlighting the importance of investing in initiatives that create a supportive and conducive environment for inter-professional learning. Previous studies pointed out that unclear role positioning was an obstacle to inter-professional cooperation (*Wei et al., 2022*). It is suggested that in the future, nursing managers should strengthen the recognition of the role and responsibilities of WOC nurses to promote the effective development of IPC.

### Serving as a specialty team leader

Our study indicated that serving as a specialty team leader was associated with a higher level of IPC competency among WOC nurses, with a small effect size ($\beta = 0.12$, $P < 0.008$). Despite a small effect size, the association between team leader and IPC competency still carries significant clinical significance. Team leaders play a crucial role in fostering a positive environment for collaboration and promoting IPC competencies among nurses. Even subtle influences from leaders, such as demonstrating collaborative behaviors or setting clear expectations, can encourage a culture of collaboration. Small, yet consistent improvements in leadership can collectively contribute to a significant overall impact on IPC improvement. This finding also aligns with international studies that highlight the critical role of leadership in IPC competency development across cultures (*Folkman, Tveit & Sverdrup, 2019*; *Lateef, 2018*). In China, clinical nurses who serve as specialty team leaders are responsible for quality management, teaching, and research, acting as leaders

in their areas of expertise (*Yang, 2021*). Studies suggest that team leaders engaged in quality management and clinical teaching are more likely to focus on their professional development and are more motivated to enhance their skills and knowledge through learning and exchange activities (*Chen et al., 2024*). When dealing with complex chronic wounds, team leaders are more aware of the strengths of different professions and adopt a more positive attitude toward IPC. They also emphasize patient-centered approaches to develop comprehensive and effective wound treatment plans. Managing chronic wounds requires the close collaboration of inter-professional teams to formulate standardized, individualized, and continuous comprehensive treatment plans (*Moore et al., 2019*). Therefore, nursing managers should emphasize the role of specialty group leaders in inspiring other WOC nurses to engage in IPC. They should organize IPC training sessions, share experiences, and establish diverse specialty positions to empower nurses in different roles, thereby enhancing their participation in IPC.

The study has several limitations. First, all participants were selected from the National Wound Care Nurse WeChat Group using a convenience sampling method, potentially introducing sampling bias and limiting the generalization of the results. WeChat groups are often formed around specific professional or social networks, and convenience sampling *via* WeChat groups may lead to overrepresentation of urban nurses and underrepresentation of rural/community-based nurses. In addition, most WOC nurses in the WeChat group worked in higher-level hospitals with higher education and skills. Therefore, our results may overestimate the level of IPC competency of WOC nurses in China. To reach a broader range of nurses, researchers should consider using multiple recruitment channels beyond a single WeChat group, particularly those targeted at rural or community-based nurses.

Second, response incentives or possible motivation for participation may also affect the study's representativeness. As participants were recruited from a professional WeChat group, those with higher levels of IPC may likely have higher motivation to participate in the survey, potentially inflating IPC scores. Future studies should carefully consider participant motivations, strategically use appropriate incentives, and prioritize ethical research practices to increase response rates and improve the representativeness of survey data.

Third, the cross-sectional study design precludes causal inferences between IPC and associated factors. In addition, the cross-sectional design may not accurately reflect the incidence of a condition and cannot track its changes over time since all data were collected at a single point in time. Future longitudinal study designs are needed to get a comprehensive overview of the study variables and establish causal relationships among them.

Fourth, all data were collected based on self-report, which may lead to inaccurate or incomplete information in questionnaire responses. For instance, participants may over-evaluate their levels of IPC due to the social desirability bias, where respondents tend to answer questions in a manner that will be viewed favorably by society. To minimize self-report bias, researchers can use multiple data sources, ensure anonymous reporting, carefully design questions, and employ mixed-method approaches.

Fifth, the use of an online questionnaire *via* SoJump might have introduced response bias despite efforts to filter surveys based on completion time. Future studies should carefully design surveys and potentially incorporate other data collection methods, such as face-to-face interviews, to mitigate these biases and improve the validity and generalizability of the findings.

Finally, the nursing work environment was assessed using a Chinese scale, which may limit its cross-comparison in a broader international context. Future studies should aim to improve data collection methods to derive more rigorous and accurate conclusions.

## CONCLUSIONS AND RECOMMENDATIONS

In this study, WOC nurses' IPC competency was at a reasonable level, but there is still room for improvement. Readiness for interprofessional learning has the strongest association with WOC nurses' IPC competency ($\beta = 0.46$, $P < 0.001$), followed by better work environment ($\beta = 0.39$, $P < 0.001$), and being a WOC team leader ($\beta = 0.12$, $P = 0.008$).

Based on the findings, we propose the following specific recommendations to improve WOC nurses' IPC competency. First, the work environment should be improved by fostering a collaborative and respectful culture that encourages teamwork across various disciplines. Some examples include holding regular team meetings among multiple healthcare professionals and using standardized communication tools to promote open discussion and ensure effective communication. Second, education and training programs should be developed to integrate IPC into routine medical and nursing curricula. Clear role definitions and expectations should be set up for each healthcare collaborative team member to promote their understanding and practice of IPC. Third, WOC nurses' leadership development should be encouraged by enhancing their professional autonomy and empowering them to act as leaders and jointly make decisions. These measures can improve their IPC competency, enabling them to develop high-quality and comprehensive care plans for patients.

### List of abbreviations

| | |
|---|---|
| **CICS29** | Chiba inter-professional competency scale |
| **IPC** | Inter-professional collaboration |
| **NWES** | Nursing work environment scale |
| **RIPLS** | Readiness for inter-professional learning scale |
| **WOC** | Wound, ostomy, and continence |
| **WOCN** | Wound, Ostomy, and Continence Nurse Society |

## ACKNOWLEDGEMENTS

We want to thank all the participants in this study.

### Funding

This study is funded by the Research Project of Natural Science Foundation of Hunan Province (2024JJ6690). The funders had no role in study design, data collection and analysis, decision to publish, or preparation of the manuscript.

### Grant Disclosures

The following grant information was disclosed by the authors:
Research Project of Natural Science Foundation of Hunan Province: 2024JJ6690.

### Competing Interests

The authors declare there are no competing interests.

### Author Contributions

- Lin Shi conceived and designed the experiments, analyzed the data, authored or reviewed drafts of the article, and approved the final draft.
- Liqing Yue performed the experiments, analyzed the data, prepared figures and/or tables, authored or reviewed drafts of the article, and approved the final draft.
- Xiaowan Liu performed the experiments, prepared figures and/or tables, authored or reviewed drafts of the article, and approved the final draft.
- Xueqin Gong performed the experiments, authored or reviewed drafts of the article, and approved the final draft.
- Bingfa Li conceived and designed the experiments, authored or reviewed drafts of the article, and approved the final draft.

### Human Ethics

The following information was supplied relating to ethical approvals (i.e., approving body and any reference numbers):

The Ethics Committee of Xiangya Hospital of Central South University (No. 2024121785).

### Data Availability

The raw measurements are available in the Supplementary File.

### Supplemental Information

Supplemental information for this article can be found online at http://dx.doi.org/10.7717/peerj.20006#supplemental-information.

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
