# Peer review of "Interprofessional collaboration and associated factors among wound, ostomy, and continence nurses: a cross-sectional study in China"

_PeerJ, doi:10.7717/peerj.20006_

## Round 0.1 · original submission · Minor Revisions

Reviewer 1 ·

Basic reporting

1. BASIC REPORTING
Strengths:

The manuscript is well written in clear, professional English.
The introduction provides appropriate context, with citations from both global and Chinese sources.
Figures and tables are relevant and referenced appropriately.
Ethical approval and participant consent are documented.
Areas for Improvement:

Minor grammatical inconsistencies are present (e.g., inconsistent use of spaces before punctuation, typos such as “ComC
Abstract: Replace “³=” with standard regression notation (β= or B=).
Consider tightening the introduction (~lines 34–84) by avoiding redundancy in definitions (IPC is defined twice) and streamlining references.

Experimental design

2. EXPERIMENTAL DESIGN
Strengths: Study design aligns with the aim: a cross-sectional survey is suitable for assessing IPC competencies and associated factors.
Inclusion/exclusion criteria are appropriate and clearly stated.
Validated tools (CICS29, RIPLS, NWES) are used with reported Cronbach's α.
Limitations:
Sampling bias: Convenience sampling via WeChat group likely excluded rural/community-based WOC nurses.
The sample may be skewed toward more educated, tertiary-hospital-based nurses, potentially inflating IPC scores.
Consider discussing response incentives or possible motivation for participation to understand representativeness.
Add a power calculation to justify the sample size of 247.

Validity of the findings

3. VALIDITY OF FINDINGS
Statistical Methods Review:
Descriptive Statistics:

Means, SDs, and proportions are appropriately reported.
Inferential Statistics:

T-tests and ANOVA: Appropriately used for comparing IPC scores across categorical variables.
However, p-values are inconsistently reported. E.g., gender comparison has t = -8.004, p = 0.424. This high t-value is incongruent with a nonsignificant p-value and warrants rechecking.
Pearson correlations: Appropriately used for continuous variables, and values reported make sense.
Multiple linear regression:
Model explains 55.1% of variance in IPC scores—this is substantial.
Assumptions (e.g., multicollinearity) are addressed (VIFs < 1.3).
Results are robust, though β vs. B notation is inconsistently used, and SE columns are not labelled properly in Table 4.

Additional comments

Strengths:

Discussion is generally well-structured and interprets key results clearly.
Limitations are acknowledged: sampling bias, cross-sectional design, and cultural context of instruments.
Some results (e.g., team management scores being low) are attributed solely to culture or education—consider including alternative hypotheses such as job demands or burnout.
Conclusion could be better aligned with the regression findings. For example, emphasize that readiness for interprofessional learning has the strongest association (β = 0.459).

Clarify whether effect sizes (e.g., β values) have clinical or practical significance.
Consider expanding the limitations: self-report bias, cross-sectional limitations beyond causality, and possible social desirability effects.

With attention to statistical consistency, clearer labelling in tables, and more robust limitation discussion, the manuscript will offer meaningful insights into WOC nursing practice and IPC education.

·

Basic reporting

The manuscript is well-written and structured, with clear academic language and logical flow. It provides relevant context, highlights a knowledge gap in IPC among Chinese WOC nurses, and uses appropriate literature and validated tools. Overall, the study is self-contained, and its findings are clearly interpretable.

Experimental design

This study is original and relevant. It addresses IPC competency among Chinese WOC nurses using a well-defined research question, validated tools, and appropriate statistical methods. The cross-sectional design was proper, although sampling was practical, and this was acknowledged. Ethical approval, clear criteria, and detailed methods support the credibility and replicability of the study.

Validity of the findings

This study presents data with appropriate statistical analysis, including regression modeling. Potential biases from sampling and self-reporting are acknowledged, and the sample size ensures valid power. The data supports the conclusions and provides practical recommendations. Limitations are discussed transparently, enhancing the credibility of the study.

Additional comments

The manuscript states that IPC among WOC nurses in China is at a “good” level (scoring rate: 92.4%). However, the comparative data used are Indonesia, Ethiopia, Italy, and Egypt, which have very different contexts of health systems and nurse professionalization. Therefore, the author needs to add comparisons from East Asian countries, such as Korea, Taiwan, Hong Kong, and Japan, which are more similar in their cultural and organizational systems.

The discussion section is too long, mixing many ideas in one enormous paragraph (pp. 12–17).

Factors discussed in the discussion are not only those with the highest scores, but also the factors with the lowest scores need to be discussed; for example, team management skills in CICS29, Basic security in NWES, and Role and responsibility of RIPLS

There is a difference in data between table 2 and the score and p-value on lines 225-228.

This article uses 3 instruments to measure the IPC scale. It is necessary to explain in detail the value of the IPC competencies described in the discussion session on line 250. Could you please clarify the reference scoring rate of 92.4% and indicate where this value is documented?

It is necessary to make metadata corrections to references. For example, in line 440 the author's name is JW. G. Should Goldsberry, JW, and in line 484 only write the name of the author X.H.?

---

## Round 0.2 · accepted · Accept

We acknowledge that the revisions you submitted in response to the reviewers' comments were appropriate and thoroughly addressed all concerns raised during the peer review process. Based on the scientific merit of your study and the quality of your responses, we are confident that your manuscript meets the standards of PeerJ and is suitable for publication.

·

Basic reporting

-

Experimental design

-

Validity of the findings

-

Additional comments

Thank you for considering all comments.